# Analysis of Gastrointestinal Acoustic Activity Using Deep Neural Networks

**DOI:** 10.3390/s21227602

**Published:** 2021-11-16

**Authors:** Jakub Ficek, Kacper Radzikowski, Jan Krzysztof Nowak, Osamu Yoshie, Jaroslaw Walkowiak, Robert Nowak

**Affiliations:** 1Institute of Computer Science, Warsaw University of Technology, 00-665 Warsaw, Poland; jakub.ficek.stud@pw.edu.pl (J.F.); k.radzikowski@ii.pw.edu.pl (K.R.); 2Graduate School of Information, Production and Systems, Waseda University, Tokyo 169-8050, Japan; yoshie@waseda.jp; 3Department of Pediatric Gastroenterology and Metabolic Diseases, Poznan University of Medical Sciences, 60-572 Poznan, Poland; jan.nowak@ump.edu.pl (J.K.N.); jarwalk@ump.edu.pl (J.W.)

**Keywords:** sound analysis, bowel sounds, gastroenterology, machine learning, neural network, deep learning, software system, spectrogram

## Abstract

Automated bowel sound (BS) analysis methods were already well developed by the early 2000s. Accuracy of ~90% had been achieved by several teams using various analytical approaches. Clinical research on BS had revealed their high potential in the non-invasive investigation of irritable bowel syndrome to study gastrointestinal motility and in a surgical setting. This article proposes a novel methodology for the analysis of BS using hybrid convolutional and recursive neural networks. It is one of the first methods of using deep learning to be widely explored. We have developed an experimental pipeline and evaluated our results with a new dataset collected using a device with a dedicated contact microphone. Data have been collected at night-time, which is the most interesting period from a neurogastroenterological point of view. Previous works had ignored this period and instead kept brief records only during the day. Our algorithm can detect bowel sounds with an accuracy >93%. Moreover, we have achieved a very high specificity (>97%), crucial in diagnosis. The results have been checked with a medical professional, and they successfully support clinical diagnosis. We have developed a client-server system allowing medical practitioners to upload the recordings from their patients and have them analyzed online. This system is available online. Although BS research is technologically mature, it still lacks a uniform methodology, an international forum for discussion, and an open platform for data exchange, and therefore it is not commonly used. Our server could provide a starting point for establishing a common framework in BS research.

## 1. Introduction

The gastroenterologist’s basic tools, including medical history, physical examination, laboratory tests, endoscopic examinations and standard imaging, complement several expensive and usually invasive specialist investigations. They include, among others, methods of examining bowel function (motility) such as multiple X-rays following oral tracer administration or rectal infusion of contrast media, as well as antroduodenal or colonic manometry, requiring general anesthesia before insertion of the probe. These methods have several disadvantages, such as a high degree of invasiveness (radiation, anesthesia, and catheter insertion), financial and time costs, and in many cases, very poor availability. Above all, they do not assess the motility of the small intestine, apart from at a few specialist centers. There is a need for a new, non-invasive method to assess the motor function of the intestine.

We propose analyzing sounds made by the digestive system, called bowel sounds (BS), by computer programs (software). BS analysis has greatly developed since 1958, when the first recordings were described [1]. First computerized analyses of BS were conducted in 1970s [2], however technical solutions have still not been made available to physicians, despite technological progress.

One of the key studies that best demonstrated the clinical potential of BS research was published by Craine et al. in 1999 and discussed differences in characteristics of BS patterns between patients with irritable bowel syndrome and controls [3]. Furthermore, at this time, stationary–nonstationary filters with wavelet transformation were introduced [4], later enriched with fractal dimension analysis. Wavelet transformations have also been used with Wiener filtering [5]. Principal component analysis of BS was done in early 2000s, along with considerable work on identification of anatomical location of BS sources [6]. Much research has also been done on signal denoising using various filters, mostly based on wavelet transformations or frequency thresholds [7]. Importantly, Kim et al. who reported BS detection with back-propagation neural networks [8] also correlated BS with colonic transit time. Real-time BS detection was achieved using a naive Bayesian classifier [9]. Support vector machines have also been used for BS identification [10], including a wearable device [11]. Other more recent approaches included spectral entropy analysis, Gaussian Hamming distance analysis [12]. In China, Liu et al. applied voice recognition neural networks to BS identification and obtained encouraging results [13]. Much progress was achieved by the Noisy Guts team from Australia. They have not only proposed their own neural network-based approach to BS identification, but also studied irritable bowel syndrome and demonstrated a link between BS and migrating motor complexes in the large bowel [14]. For more historical context we refer to our recent review [15].

The Noisy Guts team [14], who have also studied intestinal sounds, distinguished five types of sounds based on 20 h of recordings: short sounds (single bursts), multiple sounds (multiple bursts), long irregular sounds (random continued sounds), harmonic sounds (harmonic sounds) and a combination of the sounds previously mentioned (combination sounds). This large variety of detected sounds strongly suggests that there is still room for improvement in BS analysis.

The division mentioned above does not result from various mechanisms in the human body but only from an analysis of intestinal sound recordings. As a result, characteristics of some types may easily overlap. In the case of short sounds, some of them are very quiet, and it is sometimes debatable whether a sound should be considered an intestinal sound or just be ignored. The method described later in this paper does not focus on classifying sounds into types but only on their detection. This is because it is potentially most useful in medical research to quantify intestinal sounds.

Most current solutions to detect intestinal sounds are based on the extraction of features from recordings without the use of machine learning methods [16]. One of the simplest methods is to detect a sound if the average amplitude of a sound signal in a given time interval exceeds a specific value. The disadvantage of this method is that it does not solve the problem of recording noise. A simple improvement of the method is the determination of the average amplitude solely based on components with specific frequencies. More sophisticated solutions also include noise reduction using adaptive filters [17].

Machine learning methods are widely used in bowel sound analysis [15]. One solution using machine learning algorithms is the use of the parameters *Jitter* and *Shimmer*, i.e., parameters that determine the variation of the fundamental frequency in relation to the frequency of the fundamental tone and the variation of the amplitude in relation to the average amplitude. On this basis, detailed attributes are distinguished, which constitute the input to the logistic regression model. Based on a small sample of data, this method has achieved a sensitivity of 86.3% and specificity of 91.0%.

An example of using a neural network to recognize intestinal sounds is the *AIMAS* (Autonomous Intestinal Motility Analysis System) system [18]. The first step in its algorithm is to remove noise and amplify the signal using predefined filters. The time, frequency, and signal parameters derived from wavelet transform are then inserted as inputs into the dense neural network. This method allows the detection and classification of intestinal sounds with accuracy of 94.3%. The usage of neural networks is also presented in [19].

Our approach, not explored in any depth elsewhere, is to analyze sounds made by the digestive system using machine learning models based on a particular type of deep neural network. There are solutions for recognizing intestinal sounds that use dense neural networks, but so far, there have been no methods based on convolutional or recursive networks. Due to the widespread application of these networks elsewhere, an attempt to apply them to detecting intestinal sounds is interesting and worth investigating.

We have collected data using a new device with a dedicated contact microphone. We then created a dataset with 321,000 records of length 10 ms that medical doctors labeled. Our research hypothesis is that developing a new model using a deep neural network trained on a high-quality dataset enables bowel sound detection with high accuracy, specificity and sensitivity.

## 2. Methods

### 2.1. Data Acquisition

An intestinal sound-dedicated contact microphone was designed and produced at Poznan University of Medical Sciences (Poland). It consisted of an electromagnetically (copper foil) and acoustically isolated (foam) low resonance frequency (1.3 kHz) piezoelectric transducer (Ningbo Kepo Electronics, Ningbo, Zhejiang, China) enclosed in a custom design low-profile (≈1 cm) 3D-printed head containing a standard stethoscope membrane made of epoxy resin and glass fiber. The enclosure was printed with a notch around its circumference, similar to a stethoscope head, enabling mounting of the stethoscope membrane rim. The microphone was attached to the skin using 2–3 cm of two-sided medical tape (hair system tape, low-strength). The microphones work with lightweight Tascam DR-10CH (TEAC, Tokyo, Japan) recorders that capture sound with a depth of 24 bits, making it possible to record both quiet tones and individual events that are very loud.

The following protocol was established, guaranteeing the highest sound quality: the patient was instructed not to eat large meals after 18:00 and snacks after 21:00. Before going to bed, the patient was required to stick the microphone on the lower right part of their abdomen and switch on the recorder. In the evening, the patient also had to complete a clinical and nutritional questionnaire. A morning stool sample was collected to correlate selected parameters with automatically obtained quantitative data assessing bowel function.

Apart from eliminating private conversations and other disturbances, the protocol adopted had a significant advantage from the point of view of physiology: due to frequent meals, in many patients the third phase of intestinal motor activity (migrating motor complexes) appears only in sleep. It is characterized by increased peristalsis, the purpose of which is to cleanse the small intestine. The first and the second phases occur both at night and during the day. A person is not conscious and movement is minimized. There is a high probability of a long period without food. In the morning, the activity of the autonomic nervous system changes. Our preliminary results suggest that the third phase is audible and generates a loud rhythmic signal, coupled with respiratory movements. The recordings obtained may also allow detection of sleep apnea and heart rate changes.

We currently have a collection of almost one hundred recordings created by a team from the Department of Pediatric Gastroenterology and Metabolic Diseases in Poznań, Poland. Ultimately, we intend to have the largest and most diverse collection in the world. Some of the material has been tagged by marking the start and end times of thousands of recording fragments that contain intestinal sounds of interest to us. The dataset marked in this way was used to train the above-described neural network to detect interesting sound events.

The recordings from 19 subjects were divided into 2-s fragments and then mixed. The data set consisted of 321,000 records of length 10 ms. 15% of randomly selected records were extracted as test data, with the remaining 85% used for training and fivefold cross-validation.

### 2.2. Algorithm Overview

The recording is input for the algorithm depicted in Figure 1. This algorithm reduces the detection and characterization of intestinal sounds to a binary classification of recording fragments called frames.

The conversion between the audio signal to spectrogram is performed using Discrete Fourier Transform. The spectrogram is created by calculating spectral components using the window that shifts alongside the time axis. The signal was converted from the function of time x(t) to the function of time and frequency X(t,f). The process involves using Short-Time Fourier Transform, depicted in Equation (Equation 1), where *L* is the window size, $w_{n}$ multiplier dependent on the window function, *M* is the number of frames related to the total signal length and the window shift.
(1)$$X_{k,m}=\sum_{n=0}^{L-1}w_{n}x_{mL+n}e^{-2\pi i\frac{kn}{L}},\;k\in \left\{0,\ldots ,L-1\right\}\wedge m\in \left\{0,...,M-1\right\}$$

### 2.3. Parameters Optimization

In all numerical experiments, we used binary cross-entropy as a function of cost. The algorithm ADAM [20] implemented in the Keras library with default parameters was used for optimization. The experiments examining the influence of a given parameter were carried out on one architecture in which the only factor that changed was the parameter under study.

The metrics presented later show the quality of frame classification by a given model. The data presented contains the results of the best epochs during the training. The figures with the learning curves are from a one-time training of the model lasting 100 epochs.

Due to the huge space of neural network hyperparameter searches and the high computational cost of training, we carried out only a small subset of possible experiments. For this reason, we assume that the parameters could be significantly better, and, as a result, it would be possible to obtain a better quality of the model. In particular, we did not test the influence of the optimizer and several types of augmentation used to improve the model’s quality.

For parameters optimization We used sub-set containing 114,000 records.

First, we store the predictions performed on a straightforward underlying algorithm to obtain a baseline. It consisted of determining the average value of the spectrogram elements from the 0–1500 Hz interval. The frame was then considered as a bowel sound or not by comparing its mean with a predetermined threshold value. The results are shown in Table 1. The highest accuracy achieved was 86.55%, with a sensitivity of 13.83%. The base model shows that bowel sounds are mostly quiet compared to the noise in the recording. Algorithms based on straightforward attributes extracted from the recording are poorly suited to detecting these sounds.

### 2.4. Spectrogram Frequency Range

The maximum frequency determines the highest frequency of the spectrogram entering the neural network. The goal is to find a frequency above which sounds are sporadic because allowing scarcer sounds to be part of the main group makes it difficult for the model to do its job.

Based on the results of Table 2, it can be seen that the best value for the maximum frequency is 1500 Hz. Further increasing the range does not improve the results and even worsens them slightly, although the differences are not great, as can be seen in Figure 2. This is consistent with the fact that the range of short sounds is 60–1500 Hz. The biggest drop in model quality is seen for *max freq* equal to 500 Hz. This proves that the frequency range in the 500–1000 Hz range is important for prediction and intestinal sounds are often visible in this range.

### 2.5. Frame Width and Frame Smoothing Influence

The smaller the width of the *fft* window, the greater the resolution of the time domain spectrogram. However, its increase comes at the cost of reducing the resolution of the frequency domain. Intuitively, there should be a window width for which the resulting spectrograms perform best as inputs to the neural network. For each width change, the *hop length* window step was changed as follows $\left\lfloor \frac{fft}{4}\right\rfloor$. Such a step value serves to smooth and increase the frame resolution.

Based on the results from Table 3 and the Figure 3, it can be concluded that by far, the best value of the *fft* window width is 441 samples, which coincides with the observation that for this value, intestinal sounds are best seen on the spectrogram.

Two kinds of window functions were tested: Hann (Equation 2) and Hamming (Equation 3) showed on Figure 4.
(2)$$w_{Hann}\left(n\right)=0,5\left(1-cos\left(\frac{2\pi n}{N}\right)\right)$$
(3)$$w_{Hamming}\left(n\right)=0,54-0,46cos\left(\frac{2\pi n}{N}\right)$$

The results are presented in Table 4. It shows that using the Hann window clearly gives better results. In this case, the value chosen on the basis of the best visibility in the spectrogram also turned out to be the best.

### 2.6. Number of Frames for Analysis

The study of the effect of the number of frames was to find the optimal number of contiguous frames used for prediction in the linking model. Intuitively, one 10 ms frame is not enough to classify. The average duration of the briefest short intestinal sound is approximately 20–30 ms. Analysis of the intestinal sound environment on a spectrogram, which is not visible from one frame, is also useful in prediction. On the other hand, too many frames lead to an increase in the number of model parameters and make the training process more difficult.

The results are shown in Table 5 and Figure 5. It can be seen that for a small number of frames, the quality of the model is clearly worse. Above 7 frames, the difference becomes less and less noticeable. With 13 frames, the accuracy of the model reaches the highest value. Despite the best result obtained for 13 frames, the number of 9 contiguous frames was selected as the best model due to the smaller number of training parameters of the model and only slightly worse results, when compared to 13 frames.

### 2.7. Spectrogram Creation

Based on the research conducted, the best model hyperparameters were determined. The following parameters were selected to create the spectrogram:window width: 441 samples,window step: 110 samples (25%),frequency range: 0–1500 Hz,window smoothing function: Hann window.

The classified frame width is 10 ms (441 samples). This value was adopted because the frame width should be as small as possible. Otherwise, the problem is caused by having several intestinal sounds in one frame or adjacent frames with two different intestinal sounds wrongly identified as one sound.

A frame is classified as an intestinal sound if either of the two conditions is met. The first is when the duration of the intestinal sound in the frame is longer than half the frame width. The second is when more than half the duration of the entire intestinal sound is in a frame. Due to the short frame width, the second condition is met very rarely but allows the potential use of the algorithm for a larger frame size in the future.

The algorithm uses decibel spectrograms. Due to the low frequencies of intestinal sounds, we used a linear scale—there was no point in using the Mel frequency scale. The spectrogram was standardized before entering the network, based on the mean and variance of decibel amplitudes determined from all available recordings.

### 2.8. Classification

Machine learning methods detected sound events in the solution presented. The artificial neural network detected signals with specific acoustic characteristics by analyzing the spectrogram obtained from recordings. Our model involved a convolutional network (CNN) that played the role of extracting features from the images (spectrograms). We chose two neural network architectures. Both used the ability of convolutional networks to generate well generalizing activation maps. The first model was a combination of the CNN and a recursive network (RNN), and the second model was a combination of the CNN with the dense neural network. The recursive part and dense part operated on a sequence of input data rather than on individual samples. We presnet the schema of the classifier in Figure 6 and Figure 7. In both our models, the first step was to obtain attributes representing the classified frame based on the convolution layers. Then, based on the sequence of vectors of these attributes, classification of each frame was performed. After prediction by the net, adjacent frames considered to be a single bowel sound were combined and treated as the same longer sound. The exact model architectures (layers description, number of neurons, etc.) along with learning hyperparameters are described on the project web-page.

Note that we examined and experimented with a wide variety of classifiers, ranging from more traditional approaches like decision trees and SVM, naive Bayesian to state-of-the-art deep learning classifiers. We fine-tuned the parameters and meta parameters of our algorithms during the experiments. The details and results are available on the project webpage.

#### 2.8.1. Convolutional and Recursive Models

The first model described is a convolutional model with a recursive model called the CRNN [21] model. It works by extracting graphical information from the sequence of vectors obtained, thanks to the convolutional network information about the class using a recursive network. The network diagram is shown in Figure 6. The solution uses convolution layers with two-dimensional filters and max pooling layers to extract the attributes from the frame. For sequence analysis, the bidirectional gated recurrent units layer (*GRUs*) is used to return the sequence. The dropout layers are used to regularize the network. We used the ReLU activation function at the output of each convolutional and recursive layer. The output was the probability value of being a bowel sound, calculated for a given frame. The output was obtained using a sigmoid.

CRNN has 342,000 parameters, it is composed of the following:Input layer with dimensions: 200 (sequence length) × 15 (spectrogram height) × (width) [(Input(200, 15, 4)]2D Convolution layer with dimensions: 30 (number of filters) × 3 × 3 (filter size) and *ReLU* activation function [Conv2D(30, 3, 3, ReLU)]2D Convolution layer with dimensions: 60 × 4 × 2 and *ReLU* activation function [Conv2D(60, 4, 2, ReLU)]Layer that flattens the input attributes, yielding a tensor with dimensions of 200 × 60 [(Flatten(200, 600)]Dropout layer with 0.4 dropout probability [Dropout(0.4)]Bidirectional *GRUs* layer with 80 neurons and *ReLU* activation function [Bidirectional(GRU, 80, ReLU)]Dropout layer with 0.4 dropout probability [Dropout(0.4)]Fully connected (dense) layer with 1 neuron and *sigmoid* activation function [Dense(1, sigmoid)]Output layer with 200 × 1 dimensions

#### 2.8.2. Convolutional and Dense Models

The second model described is a model in which the classification of each frame was based on the combined attribute vectors from several frames. For example, when combining seven frames, the frame was classified based on the actual classified frame, plus three frames on the left and three frames to the right of the classified frame. In this way, it was possible to gain a broader context during classification, without using recursive networks that are difficult to train. As with the recursive model, convolution layers and max pooling layers were used to extract the attribute vector from the frame. The network diagram is shown in Figure 7.

CDNN has 115,000 training parameters, and is a build of the following specifications:Input layer with dimensions: 200 (sequence length) × 15 (spectrogram height) × (width) [(Input(200, 15, 4)]Layer that flattens the input attributes, yielding a tensor with dimensions of 200 × 60 [(Flatten(200, 60)]Fully connected (dense) layer with 120 neurons and *ReLU* activation function [Dense(120, ReLU)]Layer concatenating 9 adjacent vectors [Concat(9)]Dropout layer with 0.4 dropout probability [Dropout(0.4)]Fully connected (dense) layer with 100 neurons and *ReLU* activation function [Dense(100, ReLU)]Fully connected (dense) layer with 1 neuron and *sigmoid* activation function [Dense(1, sigmoid)]Output layer with 200 × 1 dimensions

### 2.9. Data Augmentation

The purpose of this study was to investigate the effect of data augmentation on model quality. Data augmentation is a method of artificially increasing the amount of training data available. It involves generating new data by modifying the existing data, using techniques such as image shifting or adding noise. Due to the specifics of the spectrogram and the very narrow width of a single frame, augmentation by sliding is difficult to apply. For this reason, data augmentation was performed by adding noise to the spectrogram. We used noise with normal distribution.

The results are presented in Table 6. In the case of too much noise, the sounds are jammed. As a result, augmentation has the opposite effect, as the task of recognizing intestinal sound becomes too difficult for the web. Sufficiently low noise allows improving the results slightly. However, it is debatable whether the difference is large enough to make the use of augmentation in this form profitable. The problem with noise augmentation seems to be that some intestinal sounds are very faint, so adding noise makes them virtually impossible to detect and does not help train the net.

### 2.10. Software Development Details

Software was written in Python version 3.8 and consists of 964 lines of code. We used additional libraries: librosa for preprocessing audio and TensorFlow for training models. We trained the network, ran all tests and performed numerical experiments on a PC with GTX 1080 Ti GPU (4GB RAM) and Ubuntu 20.04 64bit operating system.

The client-server system to analyse WAV recordings online is composed of 3 main components: analysis model described above, back-end server implemented in Python and Flask and front-end website designed with HTML and JavaScript.

## 3. Results

In Table 7, we present results for two machine learning models: a convolutional model with a recursive model called the CRNN [21], and a model combining a convolutional network and dense network, called the CDNN. The detailed description is given in the Methods section.

The results on testing data are presented in Table 8. These results use whole dataset, 321,000 records from 19 subjects.

We also obtained an annotated recording from another patients. This recording was used to additional validation. We used models depicted previously. The ratio of the gut sound frames to all frames in this recording is only 2.46%.

The results are presented in Table 9. The large decrease in precision is understandable due to fewer recorded intestinal sounds compared to the recordings on which the models were trained. The annotated sounds were clear, enabling the models to be more sensitive. At the same time, after manual analysis of the result files, it seems that a large number of *false positive* errors came from the inaccurate determination of frames marking the beginning and end of a sound and not from the model’s failure to detect the sound itself. The other problem with both models was the occasional treatment of a single intestinal sound as two different ones, resulting in the classification of the signal between as a noise.

We performed additional analysis on a different set of one-minute, non-annotated recordings. The CRNN model detected 146 sounds and the CDNN model 121 sounds; 34 were detected only by the CRNN model, and 9 were detected only by the CDNN model. Based on a review of the entire file, we noticed that most of the predictions were identical in both models, and that the most common differences were the shift of the beginning or ending of the sound.

The recursive model is bigger and performed better. In particular, the greater sensitivity of the combining model while maintaining similar precision is noticeable. This observation is also evidenced by the greater number of sounds found by the recursive model on an unannotated recording. In our opinion, the recursive layers are better for analyzing sequences. In addition, the recursive model can analyze the entire sequence and the model that connects only a fragment of it, which can be crucial for long types of intestinal sounds.

We developed the system, allowing medical doctors to upload and analyze samples with recorded intestine activity from patients. This tool provides a real-time analysis of the recording and statistics related to short sounds and graphical visualization of detected sounds overlaid on the recording spectrogram. In particular, the results include recording length, number of bowel sounds identified, number of bowel sounds per minute, frequency analysis in three-minute periods (mean, median, quadrille of bowel sounds per minute), duration analysis depicted individual bowel sound (histogram, percentage of bowel sounds of length 10 ms, 20 ms, 30 ms, etc.), power analysis (root mean square of the successive differences, the standard deviation of bowel sound intervals), Porta’s index, Guzik’s index, a plot of number bowel sounds per minute vs. time, a plot of the histogram of bowel sounds per minute and a plot of the histogram of bowel sound duration.

This system uses a client-server architecture. The user uploads a recording in WAV format using web-page http://bowelsound.ii.pw.edu.pl (accessed on 13 November 2021), then the analysis results are presented.

## 4. Discussion

A comparison of the results obtained with the results from other methods of detecting intestinal sounds, described in Section 1, was difficult due to the lack of a generally available set of recordings and different definitions of intestinal sounds. We present results provided by authors in Table 10. However, based on the experiments, it can be concluded that the detection of intestinal sounds by analyzing spectrograms using convolutional and recursive neural networks is possible, provides useful results and is therefore worth further research.

## 5. Conclusions

The proposed design aims to provide a solution for the analysis (classification) of BS, which would first be implemented for research purposes and then in practice. Our hypothesis states that it is possible to develop such an algorithm and employ it as a classifier of intestinal sounds.

As presented in Section 3, our algorithm achieved 97.7% accuracy, 99.0% specificity, and 77.3% sensitivity with new, high-quality labeled dataset of 321,000 records.

The methods of automatic analysis of intestinal sounds developed so far have used many different algorithmic approaches. A common feature was high efficiency but, unfortunately, a surprising lack of broader application in practice. The lack of application was due to various reasons. Practical microphones and recorders, as well as practical clinical protocols, were lacking. As our experience shows, this simple problem is a significant challenge that can only be overcome through interdisciplinary cooperation. Past researchers have systematically ignored the night-time period, which is the most interesting from a neurogastroenterological point of view. Instead, they have kept short records during the day or the first meal in the morning. The research goals in the past were preliminary and, therefore, sometimes abstract and in need of a broader application perspective and motor context. However, there is no doubt that automatic identification and analysis of intestinal sounds will be used in practice.

The main advantage of the presented approach is high quality in terms of accuracy, specificity and sensitivity. Moreover, we eliminate the problem of recording privacy (no conversations, enables data sharing). Long overnight recordings that are more likely to capture phase III of the migrating motor complex and are less influenced by eating habits, and the minimalization of external noise otherwise resulting from the daytime activity. The principal limitations of this scheme involve single sensor recordings, no information on the reaction of the gastrointestinal system to the first meal, and the lack of wireless capabilities. Nevertheless, at little cost and entirely non-invasively, highly useful information on BS may be obtained. The analysis using neural networks is computationally more expensive than some basic approaches used in the past. Still, the technology is advancing, and our model is easily deployed on new edge devices. Moreover, rapid analysis of the obtained recording should be achievable with a typical personal computer.

Our tool also has an interface designed for quick and easy tagging of the location of detected sounds. If connected to the recording device, we believe that the tool could provide a significant amount of important information related to the patient’s digestive system in real-time. Without a doubt, it will encourage more medical practitioners to record and tag their patients’ intestine activity, which, in turn, would enlarge the training dataset for our algorithm.

## Figures and Tables

**Figure 1 sensors-21-07602-f001:**
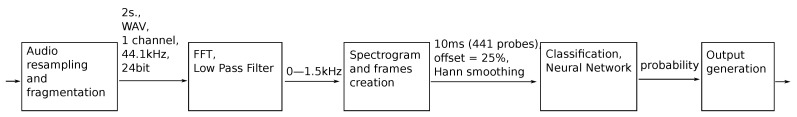
Flowchart of the method developed. The first optional step of the algorithm is converting audio into a single channel WAV format with 24 bit, 44.1 kHz sampling and represent it as a collection of 2-s recordings. Next, we remove the sounds outside interesting spectra using Fast Fourier Transform and Low Pass Filter. The signal is then converted into a spectrogram and the spectrogram obtained is divided into a sequence of frames. Finally, every frame (spectrogram) is classified using machine learning algorithms, with the returning binary value representing the frame as either bowel sound or noise. We calculate the output from adjacent frames.

**Figure 2 sensors-21-07602-f002:**
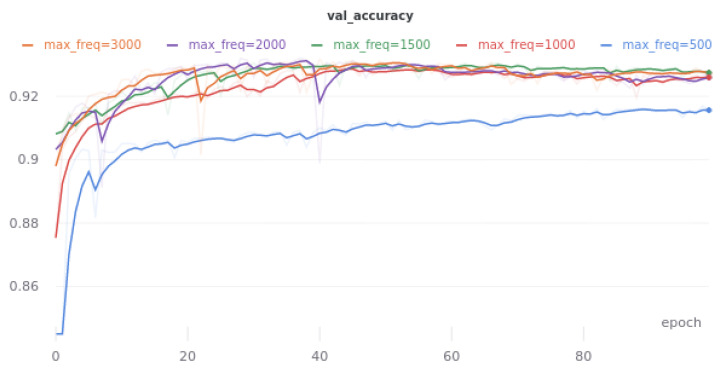
Smoothed learning curves representing the accuracy of the linking model on the validation set, depending on the maximum frequency of the spectrogram *max_freq*.

**Figure 3 sensors-21-07602-f003:**
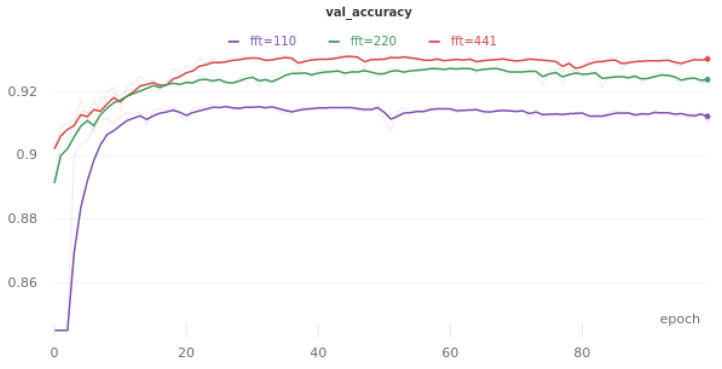
Smoothed learning curves representing the accuracy of the linking model on the validation set, depending on the window width *fft* for 3 different values.

**Figure 4 sensors-21-07602-f004:**
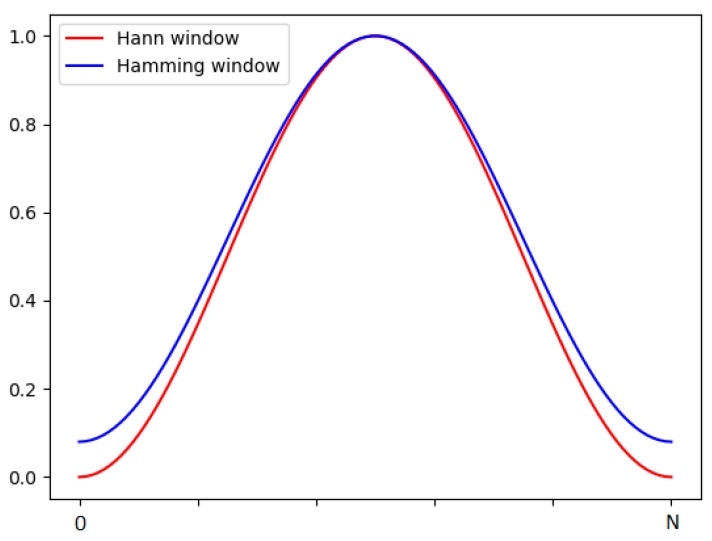
Hann and Hamming window. The x-axis represents a number of sample *n* in signal.

**Figure 5 sensors-21-07602-f005:**
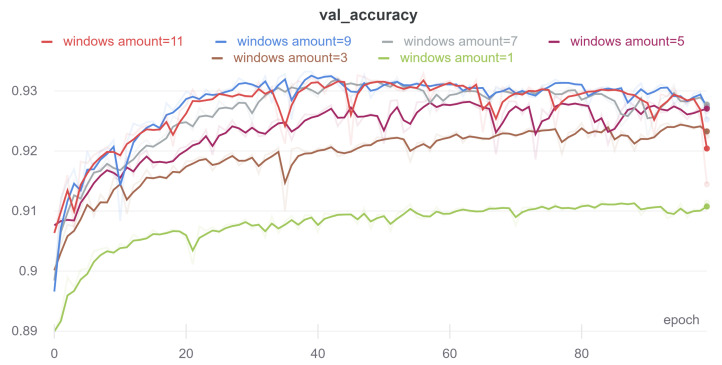
Smoothed learning curves showing the accuracy of the connecting model on the validation set, depending on the number of frames.

**Figure 6 sensors-21-07602-f006:**
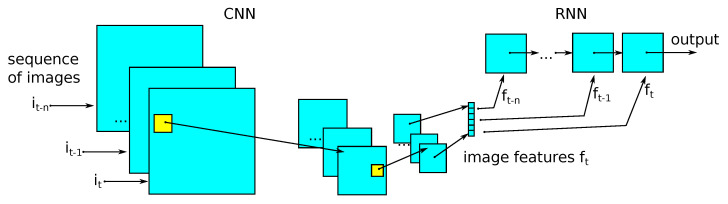
CRNN model, combination of convolutional layers (a feature extractors) and recursive layers.

**Figure 7 sensors-21-07602-f007:**
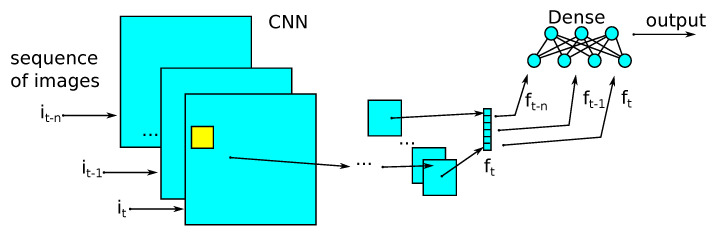
Classification model combining a convolutional network and dense network, to classify using adjacent frames results, called the CDNN.

**Table 1 sensors-21-07602-t001:** Base algorithm results. The cut-off value is the mean value of the amplitudes of the standardized spectrogram, above which the algorithm considers the frame a bowel sound.

Cut-Off Value	ACC [%]	Prec. [%]	Sensitivity [%]	Specificity [%]
0.20	85.20	97.88	0.02	99.99
0.10	86.13	94.13	0.08	99.90
0.08	86.46	88.39	11.88	99.72
**0.07**	**86.55**	**82.49**	**13.83**	**99.48**
0.06	86.47	73.51	16.27	98.96
0.05	86.12	62.91	19.61	97.94

**Table 2 sensors-21-07602-t002:** Results of the linking model on the validation set, depending on the maximum frequency of the spectrogram *max freq*. The data come from the best eras. They were obtained by the fivefold cross-validation method.

Max Freq [Hz]	ACC [%]	Prec. [%]	Sensitivity [%]	Specificity [%]
500	90.92	83.02	50.04	98.18
1000	92.69	86.21	61.38	98.25
**1500**	**92.96**	**86.55**	**63.19**	**98.25**
2000	92.93	87.19	62.34	98.37
3000	92.79	85.71	62.74	98.14

**Table 3 sensors-21-07602-t003:** The results of the link model on the validation set, depending on the window width *fft*. The data is from the best of the era. They were obtained by the fivefold cross-validation method.

fft	ACC [%]	Prec. [%]	Sensitivity [%]	Specificity [%]
1764	91.84	84.50	56.21	98.17
882	92.50	86.93	59.24	98.42
**441**	**92.92**	**85.72**	**63.66**	**98.11**
220	92.33	86.29	58.49	98.35
110	91.61	84.57	54.29	98.24

**Table 4 sensors-21-07602-t004:** The results of the linking model on the validation set, depending on the type of window function used. The data is from the best of the era. They were obtained by the fivefold cross-validation method.

Smoothing	ACC [%]	Prec. [%]	Sensitivity [%]	Specificity [%]
**Hann**	**92.99**	**87.67**	**62.32**	**98.44**
Hamming	92.78	87.08	61.83	98.39

**Table 5 sensors-21-07602-t005:** The results of the linking model on the validation set, depending on the number of frames. The data come from the best eras. They were obtained by the fivefold cross-validation method.

Num Frames	ACC [%]	Prec. [%]	Sensitivity [%]	Specificity [%]
1	91.07	86.45	48.38	98.65
3	92.06	86.30	56.37	98.41
5	92.61	85.83	61.13	98.21
7	92.84	87.54	61.33	98.44
9	92.96	86.55	63.19	98.25
11	93.00	87.51	62.55	98.41
**13**	**93.04**	**87.08**	**63.34**	**98.33**
15	93.04	86.58	63.82	98.24

**Table 6 sensors-21-07602-t006:** Results of the linking model on the validation set, depending on the variance of Gaussian noise. From each data sequence, 5 new sequences were created by adding noise with variable standard deviation **σ**. The data comes from the best eras. They were obtained by the fivefold cross-validation method.

σ	ACC [%]	Prec. [%]	Sensitivity [%]	Specificity [%]
no	92.90	87.71	61.59	98.47
**0.01**	**92.98**	**86.05**	**63.85**	**98.16**
0.02	92.91	86.93	62.48	98.33
0.03	92.97	87.02	62.79	98.33
0.05	92.92	86.93	62.48	98.33
0.10	92.59	87.28	59.56	98.46

**Table 7 sensors-21-07602-t007:** Results of the models on the validation set, depending on the type of architecture. The data come from the best epochs. They were obtained by the fivefold cross-validation method.

Model	ACC [%]	Prec. [%]	Sensitivity [%]	Specificity [%]
CRNN	93.24	85.56	66.43	98.01
CDNN	92.97	86.55	63.21	98.26

**Table 8 sensors-21-07602-t008:** Results of the best recursive and combining model on the test set. The models were trained on a set of all 5 validation subsets. The ratio of the number of intestinal sound frames to the total number of frames in the test set is 15%.

Model	ACC [%]	Prec. [%]	Sensitivity [%]	Specificity [%]
**CRNN**	**97.7**	**82.7**	**77.3**	**99.0**
CDNN	97.4	83.3	71.1	99.1

**Table 9 sensors-21-07602-t009:** Results of additional testing on the new annotated audio recording.

Model	ACC [%]	Prec. [%]	Sensitivity [%]	Specificity [%]
**CRNN**	**98.10**	**57.57**	**85.63**	**98.41**
CDNN	97.87	54.20	84.73	98.20

**Table 10 sensors-21-07602-t010:** Comparison of our algorithm with others (in alphabetical order). We present results provided by authors.

Author	ACC	Year	Algorithm
Emoto et al. [22]	91%	2013–2018	FFT, moving average
Hadjileontiadis et al. [23]	95%	1999–2011	perceptron
Inderjeeth et al. [24]	87%	2018–2020	machine learning
Kumar et al. [25]	75%	2019	SVM
Liu et al. [13]	92%	2018–2019	CNN
Ulusar et al. [9]	94%	2014	Bayesian classifier
This	97%		CNN + RNN

## Data Availability

Dataset is freely available on Kaggle, doi:10.34740/kaggle/dsv/2789758, https://www.kaggle.com/robertnowak/bowel-sounds, accessed on 13 November 2021, under CC BY-NC 4.0 License.

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
