# Peer review of "Analysis of Gastrointestinal Acoustic Activity Using Deep Neural Networks"

_sensors, 2021, doi:10.3390/s21227602_

Round 1

Reviewer 1 Report

This article proposes a methodology for the analysis of gastrointestinal sound using hybrid convolutional and recursive neural networks. Some aspects to analyze:
1. in the introduction the authors should present more modern techniques (algorithms and neural networks) and their medical importance in this area of ​​intestinal sounds.
2. also, a better description of the equipment used in the research is necessary (it is built by them, if yes, it respects the operating conditions of a medical equipment (example: compatibility tests)).
3. how many subjects were co-opted in the research (the phrase “severeal subject” appears in the text). What would be the reasoning to divide into 2-second fragments and 114 then mixed ?

4. the analysis methodology is correct, the authors emphasizing the important elements for demonstrating the principle.

5. the lack of comparison, as the authors state in the conclusions, does not make the applicability of the research very credible, but it can be concluded that the detection of intestinal sounds by analyzing spectrograms using convolutional and recursive neural networks is possible, provides useful results and is therefore worth further research.

Reviewer 2 Report

The authors have conducted interesting and modern research related to the analysis of acoustic activity using deep neural networks. They have proposed a hybrid methodology with the use of two types of artificial neural networks: convolutional and recursive. Indeed, it is a novel methodology that demonstrated promising results that have been proven by medical specialists. The authors presented numerical results also. However, in my opinion, this manuscript should be improved to reach better readability and clarity. I suggest the following comments.
1. Introduction section contains results of the study (lines 77-78). I understand the authors strive to present the advantages of their approach. Nevertheless, I am convinced the results should not be discussed in the introduction section. It is more convenient to describe the similar approaches underlying the main topic of this research in the introduction.
2. One another issue in the introduction is the lack of the aims and problems of research. I think these issues should be described more clearly. It would be interesting to the readers to compare the previous approaches and methods with the proposed one. Adding the relevant sources to the introduction section is one of the preferred cases.
3. Subsection 2.2 describes the proposed algorithm in a very condensed way. I think the description of the algorithm should be done in more detail.
4. Line 168 contains Hann and Hamming window functions. Those functions are not well-known and need to be detailed.
5. Section 4 contains the table with the results of the comparison of the proposed algorithm with others. This information, in my opinion, is more suited for the discussion section. So it would be better to add the discussion section before the conclusion to give pros and cons of the proposed approach. The conclusion section should reflect the results obtained in the main part.

Overall after improving the manuscript it can be considered as a candidate for pre-acception procedure.

Round 2

Reviewer 1 Report

Thanks for understanding the changes.

Reviewer 2 Report

I thank the authors for their patience and for improving their manuscript. I see the manuscript has been essentially enhanced and become more readable and attractive. In my opinion, this manuscript can be recommended for acceptance in the present form.